# Heterogeneous Nuclear Ribonucleoproteins Involved in the Functioning of Telomeres in Malignant Cells

**DOI:** 10.3390/ijms20030745

**Published:** 2019-02-10

**Authors:** Sergey S. Shishkin, Leonid I. Kovalev, Natalya V. Pashintseva, Marina A. Kovaleva, Ksenia Lisitskaya

**Affiliations:** Laboratory of Biomedical Research, Bach Institute of Biochemistry, Research Center of Biotechnology of the Russian Academy of Sciences, Leninsky Prospekt, 33, bld. 2, 119071 Moscow, Russia; kovalyov@inbi.ras.ru (L.I.K.); pashintseva2009@yandex.ru (N.V.P.); m1968@mail.ru (M.A.K.); lisksenia@mail.ru (K.L.)

**Keywords:** heterogeneous nuclear ribonucleoproteins, telomere, telomerase, human malignant cells

## Abstract

Heterogeneous nuclear ribonucleoproteins (hnRNPs) are structurally and functionally distinct proteins containing specific domains and motifs that enable the proteins to bind certain nucleotide sequences, particularly those found in human telomeres. In human malignant cells (HMCs), hnRNP-A1—the most studied hnRNP—is an abundant multifunctional protein that interacts with telomeric DNA and affects telomerase function. In addition, it is believed that other hnRNPs in HMCs may also be involved in the maintenance of telomere length. Accordingly, these proteins are considered possible participants in the processes associated with HMC immortalization. In our review, we discuss the results of studies on different hnRNPs that may be crucial to solving molecular oncological problems and relevant to further investigations of these proteins in HMCs.

## 1. Introduction

In 1971, Alexey Olovnikov hypothesized that with each cell division, the DNA at chromosomal ends (telomeres) is slightly shortened, and in some (cancer) cells, this shortening eliminates a special enzyme (called telomerase). Confirming this proposed hypothesis has generated considerable interest in studying telomere structure and dynamics [1,2,3,4,5,6,7]. The human telomerase holoenzyme, at a minimum, consists of reverse transcriptase (hTERT) and an RNA component (hTR) [2,3], O14746 UniProt.

By the end of the 20th century, the idea had formed that the telomeric DNA is organized into special regions of non-nucleosomal chromatin designating as telosome [8].

In the first decade of the 21st century, it was shown that human telosomes contain a six-protein complex called shelterin, which comprises TRF1 (Telomere Repeat Factor 1; P54274 UniProt), TRF2 (Telomere Repeat Factor 2; Q15554 UniProt), POT1 (Protection of Telomere 1; Q9NUX5 UniProt), RAP1 (Repressor/Activator Protein 1, Telomeric repeat-binding factor 2-interacting protein 1; Q9NYB0 UniProt), TIN2 (TERF1-interacting nuclear factor 2; Q9BSI4 UniProt), and TPP1/ACD (POT1-TIN2 Organizing Protein/Adrenocortical dysplasia protein homolog; Q96AP0 UniProt) [9,10,11]. Subsequently, some authors started to use the term “telosome/shelterin complex” [6,10,12]. On the other hand, the term “telomeres” is usually used as a broader designation of the specialized structures (nucleoprotein complexes) in chromatin that evolved to protect the ends of linear chromosomes [13,14,15].

In general, it is known that human telomeres function to support chromosome integrity and prevent DNA damage. Relevant and important processes occur in normal and pathological tissues, including malignant tumors, and more than 50 reviews have been published in 2018 alone on these issues (for example, [14,15,16]). It is believed that several peculiarities (dysregulation) in the functioning and dynamics of telomeres exist in human malignant cells (HMCs) that are not present in normal cells [1,7,17,18,19]. Moreover, it has been shown that telomerase activity is either not defined or is detectable at low levels in normal cells [20,21,22]. On the contrary, telomerase is very active and capable of lengthening telomeres in HMCs, and this activity leads to immortalization and increases the invasiveness of HMCs [7,23,24,25]. In addition, alternative mechanisms of telomere lengthening have been identified in HMCs [7,24].

Telomeres contain several nucleic acid species: double-stranded DNA repeats (5′-TTAGGG-3′/5′-CCCTAA-3′ in humans), single-stranded DNA repeats resulting from the protrusion of the 3′-G-rich strand over its complement (the G-overhang), and a single-stranded G-rich long noncoding RNA (lncRNA) known as TERRA [1,13,18,26].

TERRA is considered one of the key components of the nucleoprotein structure of telomeres, because it is involved in protecting chromosome termini from damage, as well as in maintaining chromosome length, integrity, and stability [27,28,29,30].

Over the past decade, the study of telomere structure and the role of a number of proteins, as well as various RNAs associated with telomere functions, has constituted a special research direction, as reflected by the hundreds of experimental and review publications (for example, [1,2,5,6]). Among the proteins that are capable of binding to telomeres in HMCs, several heterogeneous nuclear ribonucleoproteins (hnRNPs) have been found. Heterogeneous nuclear ribonucleoprotein A1 (hnRNP-A1) is the most studied hnRNP, and hnRNP-A1’s ability to interact with telomeric DNA and increase telomerase activity has been demonstrated [27,31,32,33]. Moreover, it has been observed that a decrease in hnRNP-A1 content leads to a decrease in telomerase activity [34,35,36]. In addition, there is evidence that individual proteins related to hnRNP-A1 (members of the hnRNP-A/B family), as well as some other hnRNPs (hnRNP-D, hnRNP-F, hnRNP-K, hnRNP-U, etc.), are able to participate in telomere stabilization [31,35,37,38,39,40].

Thus, the detected or even probable participation of individual hnRNPs in telomere function (including the maintenance of the length of telomeric DNA) in HMCs is of considerable interest in various areas of molecular oncology [35,37,39,40]. Considering the fact that such hnRNPs can be involved in various molecular mechanisms that lead to carcinogenesis, their interactions with telomeres deserve special attention.

## 2. General Properties of Several Main Human hnRNPs Involved in Telomere Functions

More than 40 years ago, Beyer et al. (1977) [41] described three groups of nuclear proteins—A, B and C—which specifically interacted with rapidly labeled nonribosomal nuclear RNA to form special heterogeneous nuclear particles—40S hnRNP particles. Accordingly, these proteins are called heterogeneous nuclear ribonucleoproteins (hnRNPs). From the mid-1980s, some authors who noted similar properties among hnRNPs, particularly immunochemical, proposed categorizing all hnRNPs into a special protein family [42,43,44]. The term “family” in relation to all hnRNPs is sometimes used today (for example, [45]). However, the identified structural and functional features of hnRNPs, especially the presence of different RNA-binding domains, have resulted in these proteins being grouped often into a special class that consists of several protein families, subfamilies, and individual members (see below).

During the 1980s, G. Dreyfuss’ group [46,47] conducted a broad, comprehensive study of hnRNPs in HMCs. Their experiments included the isolation of similar proteins from nucleoplasm by immunoprecipitation using the appropriate antibodies, as well as by affinity chromatography on single-stranded DNA-agarose and ribonucleotide homopolymers (poly(G), poly(A), poly(U), and poly(C)) immobilized on cellulose or Sepharose. As a result, these authors classified 24 proteins as hnRNPs, denoting them with Latin letters from A to U [46,47]. The proposed alphabetical classification arises from a previously used approach [41] that took into account both the nuclear localization and the electrophoretic properties of the identified proteins. 

In the last decade of the 20th century, numerous studies confirmed the existence of the overwhelming majority of the hnRNPs that were described in the 1980s, and, after deciphering the human genome, at least 30 genes encoding various hnRNPs were identified [45] and by UniProt. However, the genes for three hnRNPs (hnRNP-N, hnRNP-S, hnRNP-T) proposed by G. Dreyfuss’ group [46,47] have not yet been found in the human genome. Moreover, no other information about these proteins has been found.

As early as the 20th century, it was established that hnRNPs are RNA-binding proteins, and this property is determined by the presence of different domains and motifs in their amino acid sequences. Notably, domains called RNA recognition motif (RRM) were recognized [45,48,49].

Due to RRM and some others domains, hnRNPs are able to recognize predefined sequences of different RNAs and participate in several important molecular processes related to nucleic acid metabolism [45,50,51]. For example, the vast majority of hnRNPs are considered splicing factors, some of which are involved in nuclear export and other RNA processing events. Moreover, it has been determined that RRM domains and other special domains engage hnRNPs in alternative splicing and apoptosis regulation processes [45,52,53].

According to available data, RRM domains provide hnRNPs with the ability to bind not only RNA but also DNA [31,45,50]. Some hnRNPs, such as hnRNP-A1 with two RRMs and its synthetic analog (helix-unwinding protein, UP1), are able to form complexes with human single-stranded telomeric DNA, after that telomere functioning is changing [54,55]. Thus, it is obvious that certain domains in molecules of hnRNPs are likely to be fundamentally important for the participation of relevant hnRNPs in interactions with telomeres. It was also revealed that the vast majority of hnRNPs contain between one and four RRM domains [45,48,49].

RRMs identified in hnRNPs usually consist of 70–85 amino acid residues that form a conserved modular structure containing two consensus sequences (submotifs) that directly allow RNA binding: the octamer is designated as RNP-1, and the hexamer is RNP-2 [30,48,56,57,58]. It is believed that the RRM sequence forms a four-stranded antiparallel β-sheet and two helices packed into a β1α1β2β3α2β4 topology [30,48]. As an example, Figure 1 presents a depiction of the RRM1 domain in hnRNP-A1 with submotifs RNP-1 and RNP-2 [57] and P09651 UniProt.

### 2.1. HnRNP with Two RRM Domains

#### 2.1.1. HnRNP-A1 and Other Members of the hnRNP-A/B Family

Many hnRNPs have RRM domains located in their N-terminus [51,59], and by UniProt. For example, it was found that six human genes (*HNRNPA1, HNRNPA2B1, HNRNPA3, HNRNPA0, HNRPAB, HNRNPA1L2*) encode proteins with similar structures. These proteins are characterized by the presence of two RRM domains that are close to one another and almost equal in size (Figure 2). In vertebrates, such proteins are usually categorized as a special family—the hnRNP-A/B family [45,60].

According to the data available in UniProt, in the molecules composing the main members of the hnRNP-A/B family, the first RRM domain is slightly longer than the second (by 2–5 aa, although usually by 4 aa). As a rule, a tandem of RRM domains begins at the very edge of the N-terminal part of the molecules. An exception is hnRNP A/B, in which two RRM domains overlap by one amino acid and are located almost in the center of the amino acid sequence. Analysis of the amino acid sequences of RRM-1 domains in members of the hnRNP-A/B family showed a high degree of homology between them; as a rule, the value is 90% (Figure 2). Among the other hnRNPs containing RRM domains, only the RRM-1 domain in hnRNP-D0 (synonymous with AU-rich element RNA-binding protein 1, AUF1) has homology as high as 70% (compared with the RRM-1 domain of hnRNP-A1); in the rest, this indicator is significantly lower at 34–46%.

Moreover, the amino acid sequences of submotifs RNP-1 and RNP-2 in members of the hnRNP-A/B family are almost identical, with single amino acid substitutions occurring only in some cases (Table 1) [57] and UniProt.

It is important to note that members of the hnRNP-A/B protein family, which are encoded by five of the six above-mentioned genes, have another significant structural similarity. These proteins contain repeating amino acid triplets or RGG (glycine-arginine-rich domains) motifs in the C-terminal regions of their polypeptide sequences. This amino acid triplet is also present in the product of the sixth gene (*HNRPAB*) but only as a single copy.

The hnRNP-A1 protein has four copies of the RGG motif, all of which are located within a relatively small area (218–240 aa) and often characterized as the RNA-binding RGG-box [P09651 UniProt]. In many cases, the glycine residue is located in front of the arginine residue of RGG repeats. Consequently, the quite large, positively charged arginine radical (100 D) forms at this site (GRGG) of the polypeptide chain as a result of ionization, while the adjacent amino acid residues present only protons with a mass of 1 D instead of radicals. This structure can thus serve as the binding site for the negatively charged phosphate groups of RNA and DNA.

It is known that RGG motifs are present in the polypeptide chains of the most diverse proteins, and they, like other functionally significant regions consisting of a small number of amino acid residues (from 3 to 12), are defined as short linear motifs (SLiMs) [62,63]. It is believed that SLiMs play an important role in molecular evolution.

Given the above and a number of other structural properties, it is highly likely that the diversity of members of the hnRNP-A/B family is due to the evolution of a common ancestral gene [61,64].

The genes encoding members of the hnRNP-A/B family contain many nonsynonymous single nucleotide substitutions that lead to the formation of hnRNP isoforms (by dbSNP NCBI). In some cases, these isoforms have dramatic changes in function that can even lead to pathology in some patients, such as autosomal dominant multisystem proteinopathy or amyotrophic lateral sclerosis [65] or 164017 OMIM NCBI.

Four genes in the hnRNP-A/B family (*HNRNPA1, HNRNPA2B1, HNRNPA3, HNRPAB*) are expressed with alternative splicing [by UniProt]. So, three transcripts can form due to alternative splicing during *HNRNPA1* gene expression. The main protein product of this gene is isoform A1A with a molecular mass (Mm) of 34 kDa (P09651-2 UniProt); this is the most studied of such isoforms. This protein has been found in significant amounts in HeLa cells and other malignant tumors of epithelial or mesenchymal origin [47,53,66]. The presence of hnRNP-A1 has also been recorded for neuroblastomas [52] and gliomas [67]. In addition, in a comparative proteomic study, it was shown that the content of hnRNP-A1 in cancer cells is usually higher than in adjacent normal tissues [68]. In our laboratory, a proteomic analysis of ten cultured HMC lines revealed hnRNP-A1 as one of the 500 most abundant proteins [69].

Another isoform, hnRNP-A1B with an Mm of 38 kDa (Isoform A1-B, P09651-1 UniProt), can be synthesized in parallel with hnRNP-A1 but in smaller quantities. In HeLa cells, the content of hnRNP-A1B was estimated to be 5% compared with the content of the main isoform, hnRNP-A1A [70]. The hnRNP-A1A and A1-B isoforms are very similar in structure (Figure 2); for example, the N-terminal ends of these proteins with functionally important RRM domains are identical [P09651 UniProt]. Thus, it is possible that both isoforms can compete for binding sites on the components of telomeres.

Information on the third transcript of the *HNRNPA1* gene is extremely limited (P09651-3 UniProt). Nonetheless, recently, in our laboratory, trace amounts of the corresponding protein product in human mesenchymal stem cells (SC5-MSC) were detected by proteomic analysis [69].

A significant contribution to the diversity of hnRNPs is made by numerous post-translational modifications of these proteins. This type of processing leads to the formation of isoforms that differ from each other in their electrophoretic, chromatographic, and functional properties [52,53,66]. For example, it has been shown that the phosphorylation of Ser6 in hnRNP-A1 is accompanied by glucose metabolic reprogramming [71]. One of the consequences of structurally and functionally diverse isoforms may be changes in the effects of hnRNPs on other metabolic processes in actively proliferating cells, as well as on telomere stability [72,73]. In particular, there are experimental results that indicate that hnRNP-A1 phosphorylation is critical for capping newly replicated telomeres and preventing telomeric aberrations [73].

Accordingly, the current approach to the classification and numbering of hnRNPs should account for the pronounced structural and functional diversity of these proteins caused by both the multiplicity of their coding genes and the multiplicity of protein products that are formed during the expression of these genes. The term “proteoforms” has been proposed as a descriptor for these protein products [74,75]. Since the protein products of different genes, as well as proteoforms, can vary significantly in their properties (including their ability to interact with telomeres), it is important to assign an individual designation to each such product to prevent confusion and ambiguity when interpreting experimental results (particularly those obtained from studying HMCs).

#### 2.1.2. HnRNP-D1 and hnRNP-DL

Two related proteins—heterogeneous nuclear ribonucleoprotein D (hnRNP-D, hnRNP-D0, AUF1) and heterogeneous nuclear ribonucleoprotein D-like (hnRNP-DL, laAUF1)—contain two RRM domains and several RGG motifs [Q14103, O14979 UniProt]. 

There are a number of structural and functional differences between hnRNP-D and the members of the hnRNP-A/B family [76,77,78]. For instance, certain differences between the RRM domains of hnRNP-D and those of hnRNP-A1 and other members of the hnRNP-A/B family have been revealed (for example, [Q14103 and P09651 UniProt]). Moreover, the RRM domains of hnRNP-D serve to form complexes with AU-rich elements (AREs) in the 3′-untranslated regions mRNAs [77,78]. As a result, hnRNP-D plays a definite role in destabilizing mRNAs. However, hnRNP-D binds double- and single-stranded DNA sequences in a specific manner and functions as a transcription factor [Q14103 UniProt].

Increased levels of hnRNP-D have also been shown in gastric cancer using proteomic technologies [79].

The expression of the *HNRNPD* gene proceeds with alternative splicing, resulting in the formation of four transcripts and the synthesis of four similarly structured proteins, each containing two RRM domains Q14103 UniProt, [77,80]. Thus, the formation of different proteoforms of hnRNP-D is possible due to alternative splicing and post-translational modifications (more than 20 amino acid residues, including the phosphorylation of six serine residues) [Q14103 UniProt]. 

HnRNP-DL has been found in several HMCs (for example, HL-60 cells—a leukemia cell line) [81]; O14979 UniProt. Additional interest in hnRNP-DL has been generated by a recent report indicating that colon cancer cells contain aberrantly expressed *HNRPDL*, which promotes the growth of these cells [82].

### 2.2. HnRNPs with One RRM Domain

In contrast to the members of the human hnRNP-A/B family, hnRNP-C1/C2 isoforms (which are formed as a result of alternative splicing upon expression of the *HNRNPC* gene [83,84]) contain one RRM domain; they do not contain RGG motifs [P07910 UniProt].

Although the degree of RRM homology of hnRNP-C1/C2 isoforms with RRM-1 of hnRNP-A1 is low (~40%), these proteins are able to interact with the RNA part of telomerase (hTR) and are involved in the synthesis of telomeric repeats during DNA replication [31,85].

The study of hnRNP-C1/C2 in HMCs has been ongoing for several decades [46,83,84,85,86]. HnRNP-C1/C2 isoforms have been detected in HeLa cells [83] and other HMCs [85,86], and there is evidence that hnRNP-C1/C2 may be a biomarker of chemoresistance in gastric cancer cells [86].

### 2.3. HnRNPs with qRRM Domains

Some hnRNPs do not have typical RRM domains but contain significantly modified so-called quasi-RNA-recognition motifs (qRRMs) [87,88]. These qRRMs are noticeably smaller in size than, for example, the typical RRM in hnRNP-A1. These proteins do not contain the typical RNP-1 and RNP-2 consensus sequences but are capable of binding some nucleotide sequences [88,89]. Nevertheless, qRRM domains bind to G-tract RNA and participate in the regulation of the alternative splicing of pre-mRNAs [90,91].

Four genes in the human genome (*HNRNPH1, HNRNPH2, HNRNPH3, HNRNPF*) can encode several hnRNPs, each containing three qRRMs [by UniProt]. Three of them (*HNRNPH1, HNRNPH2, HNRNPF*) are expressed without alternative splicing, and their protein products do not contain RGG motifs. The *HNRNPH3* gene is expressed with alternative splicing, resulting in the formation of six protein products, five of which contain RGG motifs.

Some authors consider hnRNPs containing three qRRMs to be members of a ubiquitously expressed subfamily (or family) [49,89,92]. Currently, the biological significance of hnRNPs containing three qRRMs in HMCs is being studied, especially the potential role of these proteins in tumor progression [93].

There is evidence that at least hnRNP-F can bind telomeric RNA [40]. It is thus possible that other hnRNPs containing three qRRMs may participate in interactions with telomeric RNA.

### 2.4. HnRNPs with KH (the K Homology (KH) domain) Domains

Another functionally important domain that was revealed in hnRNP-K, as well as in some other hnRNPs, was described in the last decade of the 20th century [94,95]. It is called the KH domain. HnRNP-K and other hnRNPs with KH domains have the ability to bind RNA as well as ssDNA. It was shown that proteins with KH domains bind tenaciously to cytidine-rich sequences in RNA and ssDNA [94,95,96,97]. Currently, some authors consider hnRNPs containing KH domains to be important oligo(rC/dC)-binding proteins [94,95,96]. In particular, hnRNP-E1 and hnRNP-E2 contain KH domains and are characterized as two major cellular poly(rC)-binding human proteins (abbreviations: PCBP1 and PCBP2) [95]. All KH domains in hnRNPs are three-stranded antiparallel β-sheet packed against three α-helices (βααββα) [45].

Additionally, KH domains were revealed in two other Poly(rC)-binding proteins: PCBP3 and PCBP4 [45,98]. All of these proteins contain two consecutive KH domains positioned near the N-terminus and a third KH domain located at the carboxyl terminus. However, domains in PCBP3 [P57721 UniProt] and PCBP4 [P57723 UniProt] differ from the hnRNP-E1 and hnRNP-E2 domains: the domains in PCBP3 and PCBP4 are much smaller. PCBP3 and PCBP4 are cytoplasmic proteins, and this is usually the reason given for not considering them in connection with hnRNPs. It should be noted that all Poly(rC)-binding proteins are grouped into a special protein family [98,99].

KH domains have been found in different proteins associated with transcriptional and translational regulation, including RNA splicing [97,99,100,101]. Some structural and functional data show that multiple KH domains act in a combinatorial fashion to both enable the recognition of longer RNA motifs and remodel the RNA structure [100]. Recently, Zhang et al. (2016) [99] reported that the overexpression of hnRNP-E2 (PCBP2) contributes to a poor prognosis and enhanced cell growth of human hepatocellular carcinoma. Thus, the biological importance of KH domains in hnRNPs is apparent. Moreover, some data indicate that one of the hnRNPs containing KH domains is able to form a complex with a C-rich strand of human telomeric DNA [96].

### 2.5. HnRNPs with RGG/RG Domain

Finally, it should be noted that hnRNP-U, which was traditionally related to hnRNPs [102,103,104], has an amino acid sequence containing the special arginine/glycine-rich (RGG/RG) domain [105,106]; Q00839 UniProt. There are also some data on the RGG/RG domain in the structure of hnRNP-P2 (synonyms: Fused in Sarcoma protein or FUS, 75 kDa DNA-pairing protein, etc.) [107]. Due to RGG/RG domains, different proteins (particularly hnRNP-U) are capable of binding to double- and single-stranded DNA as well as different RNAs [107]; Q00839 UniProt. It is believed that hnRNPs with the RGG/RG domain are involved in several important biological processes: nuclear chromatin organization, telomere-length regulation, regulation of transcription for numerous genes, mRNA alternative splicing, and so on [107,108,109].

## 3. Human Telomeres and Their Relationship with Some hnRNPs in HMCs

The main components of telomeres—namely, telomeric nucleic acids, which are represented by double-stranded DNA repeats (dsDNA), single-stranded DNA (ssDNA), or the protrusion of the 3′-G-rich strand (the G-overhang); TERRA; and the telosome/shelterin complex—are also important participants of telomere–protein relationships [1,6,10,11,13,18,26].

It is known that telomere DNA forms specific structures—the so-called D-and T-loops—which are involved in the interactions with various proteins [13,18,110]. Moreover, telomeric G-rich DNA can form structures known as G-quadruplexes; these structures make telomeric ssDNA inaccessible to telomerase and, thus, block the telomerase reaction [111,112]. There are data reporting that telomere RNA can form G-quadruplex structures as well, and these structures are involved in intermolecular interactions with telomere DNA (DNA–RNA G-quadruplex) as well as some proteins, particularly hnRNP-A1 [113,114,115].

Telomere–protein relationships undergo complex and dynamic changes. For example, the cyclic interactions of some proteins with the main participants of telomere–protein relationships and/or different protein modifications in telomere composition have been observed [27,116,117,118,119]. Notably, it has been shown that telomerase is a principal, catalytically active component of the telomerase-associated protein machinery, but the human CST complex, which consists of three protein subunits (CTC1-STN1-TEN1), can physically interact with some components of the telosome/shelterin complex (POT1, TPP1) and acts as a terminator of telomerase activity [116]. An inventory of telomerase components in HMCs showed some imbalance in subunits (e.g., ~1150 hTR and ~500 hTERT molecules per HeLa cell), suggesting the existence of unassembled components [117]. Moreover, it is known that POT1 and telomerase both bind to telomeric ssDNA and are, in effect, competitive inhibitors of one another [118]. At last, there are recently published data describing the participation of microRNA in the binding of telomerase components and the regulation of telomerase activity [119].

Several years ago, telomere function was described as a series of orchestrated actions [27]. To date, this opinion is supported (for example, [119,120]) and many participants in this orchestra, including some hnRNPs, have been detected [12,18,35,36,40]. Some properties of the CST complex and several other participants of telomere–protein relationships in HMCs are discussed below.

### 3.1. CST Complex and Some Other Participants of Telomere–Protein Relationships

It has been established that the main participants of telomere–protein relationships in humans interact with several large protein complexes. One of them is a human heterotrimeric complex called CST [116,121,122]. Each subunit of the CST complex is encoded by its own gene. The molecular mass (Mm) of the largest subunit (CTC1, conserved telomere maintenance component 1) is estimated to be 134.6 kDa, and the two others are significantly smaller (STN1: 42.1 kDa; TEN1: 13.8 kDa) [Q2NKJ3, Q9H668, Q86WV5 UniProt]. Correspondingly, the Mm of the total CST complex may be ~190 kDa.

The human CST complex supports at least two main functions—telomere maintenance and DNA replication—through its ability to interact with the single-stranded DNA (ssDNA) of a variety of sequences [123,124,125]. It is known that human CST prefers G-rich sequences but not necessarily telomeric ones [124]. The CST complex can unfold G-quadruplex structures and thus provide a mechanism to facilitate the replication of telomeric DNA and other GC-rich regions [126]. Some data suggest that the human CST complex is a terminator of telomerase activity [116]. However, this feature was recently detailed, and it turned out that the human CST complex binds the telomeric overhang and regulates telomere length by promoting C-strand replication and inhibiting telomerase-dependent G-strand synthesis [123,127,128].

During the functioning of the CST complex, different subunits play special roles [123,129], but depletion of any of the three CST components causes steady telomere elongation in HMCs [116]. In parallel, in some model experiments, it was shown that the expression of amino-terminal (amino acids 1–701) and carboxy-terminal (amino acids 844–1217) CTC1 fragments leads to different effects in the telomerase-positive HT1080 fibrosarcoma cell line and normal primary lung fibroblasts (HLFs). In general, the expression of the amino-terminal CTC1 fragment causes a progressive reduction in telomere length in HT1080 cells, while it has no notable effect on the telomere shortening rate in HLF cells. In contrast, the carboxy-terminal CTC1 fragment promotes robust and continuous telomere elongation in HT1080 cells [116].

There is an opinion that the CST complex’s structure shares a certain level of similarity with another protein complex—heterotrimeric replication protein A (RPA) [121,122,125]. However, the overall architecture and functions of CST and RPA are distinct [126]. There is information suggesting that RPA interacts with telomeres due to its non-sequence-specific manner of binding single-stranded DNA and the importance of RPA phosphorylation in maintaining genome stability [27,130,131,132].

RPA consists of three subunits: replication protein A 70 kDa or DNA-binding subunit (RPA1); replication protein A 32 kDa subunit (RPA2); and replication protein A 14 kDa subunit (RPA3) [P27694, P15927, P35244 UniProt]. It is believed that the RPA complex involved in DNA metabolism and may also play a role in telomere maintenance by interacting with some telomere components, including shelterin [133,134,135,136]. For example, it has been shown that RPA, when interacting with single-strand telomeric DNA, acts as an antagonist to the POT1 protein and, thus, ensures DNA replication during the cell cycle while preventing the start of telomere elongation [27,133].

In addition, another protein complex was found that is involved in telomere maintenance, which is abbreviated to MRN (the protein complex consisting of Mre11, Rad50 and Nbs1 in eukaryotes) in accordance with its three MRE11–RAD50–NBS1 subunits [137,138,139]. Human Mre11 (Double-strand break repair protein MRE11A) has an Mm of 80.5 kDa; Rad50 (DNA repair protein RAD50) has an Mm of 153.8 kDa; and Nbs1 (Nibrin, Cell cycle regulatory protein p95) is 84.9 kDa (P49959, Q92878, O60934 UniProt). Accordingly, the total molecular mass of the MRN complex should be ~320 kDa. It is believed that the main function of the MRN complex is related to the provision of double-strand break repair; however, there are also a variety of data revealing its involvement in telomere maintenance [137,138,140].

Some data report that DNA-dependent serine-threonine protein kinase (DNA-PK) is also involved in telomere function [138,141]. DNA-PK is a heterotrimer that consists of DNA-dependent protein kinase catalytic subunit (Mm: ~470 kDa) and the DNA-bound Ku heterodimer (70 kDa subunit of Ku antigen, 86 kDa subunit of Ku antigen) [P78527, P12956, P13010 UniProt]. This large protein complex, which has the appropriate enzyme activity, is involved in carcinogenesis, and its inhibition sensitizes cancer cells to radiation [73,142]. It has been noted that DNA-PK phosphorylates the components of RPA and shelterin (POT1), as well as hnRNP-A1 [73,122].

Recently, several models describing the structure of telomeres in HMCs have been developed. These models are based on assumptions of the role of changes in the above-mentioned protein complexes, which help maintain the length of telomeric DNA and ensure the immortalization of these cells (for example, [143,144,145,146]). At the same time, the possibility that different hnRNPs are involved in various molecular mechanisms has been proposed in a number of studies [33,36,73,144,147]. Although ideas on the interactions between some protein components of telomeres and hnRNPs are largely assumptions, the proposed models have created prerequisites for the formation of several working hypotheses that may be tested experimentally [27,28,143].

### 3.2. Members of hnRNP-A/B Family and Telomeres of HMCs

#### 3.2.1. Isoforms of hnRNP-A1 and hnRNP-A2

In the last decade of the 20th century, data were obtained indicating that the hnRNP-A1, hnRNP-A2, and hnRNP-B1 proteins are products of the expression of two different genes (*HNRNPA1* and *HNRNPA2B1*) but have similar structures (two RRMs and four RGG motifs in each) and can interact with telomeres in HMCs [148,149]. Later, it was confirmed that it is the RRM domains that provide the interaction sites of these proteins with different RNA and DNA repeats present in telomeres [31,61,76]. So, it was shown that hnRNP-A1 and its N-terminal fragment UP1 (abbreviation for helix-unwinding protein), consisting of 195 aa and, therefore, containing both RRMs but devoid of RGG motifs, are capable of unfolding the quadruplex structure of d(TTAGGG) repeats [150]. Moreover, using the photochemical cross-linking method, Liu et al. (2017) found that the telomere RNA G-quadruplex with loops is important in the interaction between telomere RNA and hnRNPA1 [36]. Recently, it was shown that hnRNPA1 specifically recognizes the nucleotide base at the loop of the RNA G-Quadruplex [114].

It is important to point out that hnRNP-A1 is able to bind directly to TERRA [27,28,143]. In particular, Redon et al. (2013) [143] showed that hnRNP-A1 can alleviate the TERRA-mediated inhibition of telomerase after binding to TERRA. The obtained data became the basis for the creation of a three-state model for the regulation of telomerase by TERRA and hnRNPA1 [143]. Moreover, it was recently established that TERRA might accumulate in a telomere-neighboring region and bind hnRNPA1, thereby influencing hnRNP-A1 localization to the telomere [28].

HnRNP-A1 is capable of binding elements other than telomeric DNA and RNA. For instance, it was shown that this protein interacts with shelterin components (POT1, TPP1, etc.) [27,149,151]. Accordingly, there are grounds to regard these interactions as additional molecular mechanisms that ensure the preservation of the length of telomeric repeats in HMCs due to the direct and indirect effect on telomerase activity. It is possible that hnRNP-A1 is not only located in 40S heterogeneous nuclear ribonucleoprotein monoparticles but also in other structural formations (telosomes) of the nucleus, in addition to its partial residence as the free form in the nucleoplasmic pool.

In a series of experiments conducted in vitro and in vivo, it was found that the functionally important DNA-dependent serine-threonine protein kinase (DNA-PK) present in telomeres can phosphorylate hnRNP-A1 and thereby alter its ability to interact with the components of telosomes [152]. It was also noted that hnRNP-A1 phosphorylation is stimulated by the presence of DNA and hTR.

Sui et al. (2015) [73] showed that DNA-PK phosphorylation of hnRNP-A1 can act as a molecular switch, facilitating the replacement of RPA by POT1, both ssDNA-binding proteins that interact with the terminal 3′ single-stranded overhang DNA. The authors also found that failure of DNA-PK to phosphorylate hnRNP-A1 leads to a change in the operation of the “RPA-POT1 switch” and is accompanied by the induction of telomere fragility. On the basis of these results, it was concluded that DNA-PK-dependent phosphorylation of hnRNP-A1 is crucial for capping newly replicated telomeres and preventing telomeric aberrations [73].

HnRNP-A1 can act as a substrate for phosphorylation by another serine/threonine-protein kinase, VRK1 (Vaccinia-related kinase 1), which is present in both cell nuclei and the cytosol (Q99986 UniProt). There are some data showing that phosphorylation of hnRNP-A1 by VRK1 enhances its binding to telomeric ssDNA and promotes an increase in telomerase activity [153]. The ability of VRK1 to regulate the activity of hnRNP-A1 through phosphorylation is of considerable interest, especially given the fact that this enzyme is involved in carcinogenesis and is even considered a potential therapeutic target in oncology [154].

Correspondingly, when describing the participation of hnRNP-A1 in the dynamics of telomeres, it is necessary to take into account that the different proteoforms might acquire unique properties when interacting with the components of telomeres. This supposition is evidenced by data on the individual phosphorylated forms of hnRNP-A1 [73,152,153].

The expression not only of hnRNP-A1 isoforms but also hnRNP-A2 and other members of the hnRNP-A/B family (in particular, A3 and A0) was revealed in malignant cells by proteomic and transcriptomic methods [155,156,157,158,159]. For example, it was found that hnRNP A2 is an ssDNA-binding protein, and it can recognize single-stranded vertebrate telomeric repeat (TTAGGG)n [66,148]. Increased hnRNP-A2/B1 isoform content has been observed in HMCs, and these proteins interact with telomeric ssDNA and protect the telomeric DNA repeat from endonuclease digestion [160,161,162].

It was also shown that the shortened isoform, hnRNP-A2*, is capable of binding telomeric DNA and plays a positive role in unfolding telomere G-quadruplexes to enhance telomere extension by telomerase [38]. Recently, new reports have emerged suggesting that hnRNP-A2/B1 isoforms are able to bind non-coding RNA and that these complexes interact with chromatin [163]. The knockdown of hnRNP-A2/B1 inhibits cell proliferation, invasion, and the cell cycle, thereby triggering apoptosis in cervical cancer [164].

In general, hnRNP-A/B family members, which contain two RRM domains, play an important role in cell proliferation, although the significance of the functional overlap among members of hnRNP-A/B remains largely unexplained [155,165]. However, some differences between hnRNP-A1 and hnRNP-A2/B1 isoforms in HMCs were described despite their great structural and functional similarity [158,165].

So, at a minimum, hnRNP-A1 and hnRNP-A2, the most studied members of the hnRNP-A/B family, are capable of binding telomeric DNA and affect the function of telomerase [34,36,152,166]. For example, using an in vitro telomerase assay, Zhang et al. [34] revealed that depletion of hnRNP A/B proteins from 293 human embryonic kidney cell extracts dramatically reduced telomerase activity, which was fully recovered upon the addition of purified recombinant hnRNP-A1. Moreover, adding recombinant hnRNP-A2, which has 68% amino acid identity with hnRNP-A1, had a similar effect. These authors also showed (using chromatin immunoprecipitation) that hnRNP-A1 associates with human telomeres in vivo. As their conclusion, Zhang et al. [34] proposed that hnRNP-A1 stimulates telomere elongation by unwinding G-quadruplex or G-G hairpin structures. This suggestion obtained additional support later [36,166].

It should be noted that hnRNP-A1 and hnRNP-A2 are high-abundance proteins in the nucleus [167,168]. These proteins are expressed in considerable excess compared with telomerase and the number of telomeres in a human cell (46 × 2) [38,169]. In the nucleus a significant proportion of hnRNP-A1 and hnRNP-A2 is found in complexes with pre-mRNA (40S heterogeneous nuclear ribonucleoprotein monoparticles) [167,168]. However, in principle, even a small fraction of these proteins from the nucleoplasmic pool (free hnRNP-A1 and hnRNP-A2) should saturate both telomeres and telomerase. Thus, the available data suggest two hypotheses.

First, the full saturation of the hnRNP-A1/hnRNP-A2 binding sites at the terminal 3′ -stranded overhang (or in participants in telomere maintenance) might block some specific molecular regulator in HMCs. The CST complex and/or RPA possibly serve as such regulators [125,127,133]. At the same time, some data indicate that other proteins are capable of specifically binding to telomeres and affecting telomere maintenance [170,171,172,173]. Additionally, POT1, as a part of shelterin complex, binds to telomeric ssDNA and blocks telomerase function [27,118,174]. High concentrations of hnRNP-A1/hnRNP-A2 may be necessary in order to compete with such regulators. It is possible that the small sizes of hnRNP-A1/hnRNP-A2 (each are less than 40 kDa) in comparison with RPA (more than 100 kDa) and/or the CST complex (near 190 kDa) allow telomerase access to the single-stranded overhang. Moreover, these proteins might stimulate telomere elongation by unwinding G-quadruplex or G-G hairpin structures [34,166]. As a consequence, these interactions could lead to increased telomerase function.

Second, in HMCs, the regulation of telomerase activity may only be a function of some post-translationally modified hnRNP-A1 and hnRNP-A2 proteoforms. The significance of such modifications was discussed from the aspect of the regulation of the subcellular distribution of hnRNP-A1 and hnRNP-A2 [175,176]. However, it has been reported that hnRNPA1 itself has no notable direct effects on telomerase catalytic activity [143]. Nevertheless, it is impossible to completely exclude the possibility that some hnRNP-A1 and/or hnRNP-A2 proteoforms affect the activity of the enzyme by an allosteric mechanism. The potential influence on the activity of telomerase by various allosteric effectors has been noted by some authors [177,178,179]. 

The regulation of alternative splicing is important for the appearance of active hTER during *TERT* gene expression (as described, for example, in [33,180,181]). Correspondingly, some hnRNP-A1 and/or hnRNP-A2 proteoforms might contribute to such mechanisms as splicing factors.

It also is possible that both of these hypothetical mechanisms are correct. Thus, the process of inducing telomerase for the maintenance of telomeres in HMCs can be represented by several stages (steps). 

Initially, some hnRNP-A1/hnRNP-A2 proteoforms displace a hypothetical protein regulator from its complex with telomeric ssDNA.

After that, some hnRNP-A1/hnRNP-A2 (possibly others) proteoforms interact with POT1 of shelterin (which might correlate with post-translational modifications of POT1, such as phosphorylation). As a result, POT1 probably loses the ability to effectively compete with telomerase for binding to the G-overhang and thus allows telomerase to interact with the opened G-overhang. In parallel, the contribution of the hnRNP-A1/hnRNP-A2 proteoforms to the unfolding of the quadruplex structure from the d(TTAGGG) repeat might be one of the results of this step. Thus, the key roles in the initiation of telomerase activity for the maintenance of telomeres may belong to various free hnRNP-A1/hnRNP-A2 proteoforms that are present in the nucleoplasmic pool.

Figure 3 presents the corresponding hypothetical scheme of possible dynamic relationships between free hnRNP-A1/hnRNP-A2 from the nucleoplasmic pool and some components of the telosomes. The presented illustration was established on the basis of results from a number of publications [27,36,73,114,118,133,143].

The proposed scheme is not exhaustive since some well-known and proposed participants in the maintenance of telomere processes are not shown (the previously mentioned DNA-PK, CST complex, MRN complex). However, it is possible that this scheme will contribute to research that will allow us to broaden our understanding of the roles played by hnRNP-A/B family members in telomerase activation for the maintenance of telomeres.

HnRNP-A1 and some other members of the hnRNP-A/B family are present not only in HMCs but also in various normal human cells, including the cells of tissues adjacent to malignant tumors [64,68,69,182,183]. However, the content of hnRNP-A1 in normal cells (or normal tissues) is usually significantly lower compared with HMCs [68,69,159,184]. Currently, it is believed that hnRNP-A1 (and some other members of the hnRNP-A/B family) is involved in both normal and pathological RNA metabolism [50]. Nevertheless, a reduction in hnRNP-A1 and -A2 protein content by small interfering RNAs induces apoptosis in human cancer cells but not in normal mortal cell lines [184]. It might be that the members of the hnRNP-A/B family function mainly as splicing factors in normal cells [50,185] and are not able to effectively promote the maintenance of telomeres.

#### 3.2.2. HnRNP-A18

The ability to interact with telomeres was detected in another RNA-binding protein, designated as hnRNP-A18 (synonyms: cold-inducible RNA-binding protein, CIRBP, CIRP), and it is in the hnRNP-A/B family [147,186,187]. Inhibition of hnRNP-A18 activity or siRNA knockdown leads to reduced telomerase activity and shortened telomere length, suggesting an important role for hnRNP-A18 in telomere maintenance. According to some data, the content of hnRNP-A18 in HMCs is significantly increased [188]. This protein is proposed to be a potentially useful prognostic biomarker of colon cancer [189].

Unlike other members of this family, hnRNP-A18 contains only one RRM domain. However, this domain is located near the N-terminal part of the polypeptide chain (6–84 aa) and is very similar in structure to the RRM domains of other members of the hnRNP-A/B family. For example, the RRM domain in hnRNP-A18 contains the RNP-1 (RGFGFVT) and RNP-2 (LFVGGL) submotifs, which are almost identical to those found in the typical members of the hnRNP-A/B family (Table 1). In addition, hnRNP-A18 has three RGG motifs in its C-terminal region, which is typical for members of the specified family [Q14011 UniProt]. According to the available data, the expression of *CIRBP* (the gene encoding hnRNP-A18) proceeds with alternative splicing, but only isoform 1 [Q14011 UniProt] has the structural properties mentioned above.

### 3.3. Several Other hnRNPs with RRM Domains and Telomeres of HMCs

HnRNP-D can specifically bind to single-stranded d(TTAGGG)n (the human telomeric repeat) to unfold the quadruplex of this DNA upon binding and induce the maintenance of telomere elongation by telomerase [76,190,191]. Moreover, the analysis of the three-dimensional structures of the RRM domains of hnRNP-D indicates the possibility of heterodimer formation of this protein with hnRNP-A1 [78].

It has been shown that the main contents of the four hnRNP-D isoforms encoded by the *HNRNPD* gene are located in the nuclei. These proteins are considered transcriptional regulators and involved in nucleocytoplasmic shuttling, as well as other common functions [Q14103 UniProt], and this has given some authors reason to consider them as a separate family [77,80].

Apparently, due to the presence of two RRM domains in their structure, hnRNP-D proteoforms can act as competitors with members of the hnRNP-A/B family for binding to telomere components and maintaining telomeres.

Information on the interaction of hnRNP-DL with telomeres has not yet been found, but the structural similarities between hnRNP-DL and hnRNP-D is a sound basis for suggesting that this is possible.

In some publications, it was noted that hnRNP-C1/C2, with one RRM domain, can interact with telomeres and/or telomerase components. In particular, it was shown that hnRNP-C1/C2 binds directly to a six-base U-rich tract located at the 5′ terminus of the hTR template [2,31,85]. Although the deletion of these six bases from the U-rich tract in the full-length human telomerase RNA does not significantly influence telomerase activity, this deletion can abolish the ability of hnRNP-C1/C2 to associate with the telomerase holoenzyme. Moreover, it was detected by immunofluorescence that hnRNP-C1/C2 colocalize with telomeric binding proteins in interphase nuclei, but they were not shown binding directly to telomeric DNA [31,85]. It was suggested that hnRNP-C1/C2 might bind to telomeres through protein–protein interactions. At the same time, the question of the functional meaning of the binding of hnRNP-C1/C2 to components of telomeres and/or telomerases remains open.

There is a special interest in the study of the interaction between HMCs and hnRNP-P2 (FUS) telomeres, which contain one RRM domain and 18 repeats of the RGG motif (in the splice variant, 16 repeats of the RGG motif) [P35637 UniProt]. This interest is largely due to the fact that hnRNP-P2 and the gene encoding it have been described as an oncoprotein and pro-oncogene, respectively, by a number of authors [192,193,194]. The sequence of RRM in hnRNP-P2 varies significantly from canonical RRMs [195]. According to Bentmann et al. (2012) [196], the RRM domain of hnRNP-P2 has a minor contribution to DNA–RNA binding, whereas RGG domains with a zinc finger domain play primary roles in DNA–RNA-binding functions.

Special domains are present in the amino acid sequence of hnRNP-P2 and contain RGG motifs, and Takahama et al. (2013) showed their significance for binding to G-quadruplex DNA and RNA [197]. Recently, additional evidence has emerged that hnRNP-P2 binds to G-quadruplex telomere DNA and to G-quadruplex TERRA through these domains [115,198]. It is suggested that owing to this mechanism, hnRNP-P2 might regulate telomere length in vivo.

Information on the possible direct or indirect interaction between telomeres and hnRNPs, which contain more than two RRM domains, is still very limited [199].

### 3.4. HnRNPs without RRM Domains and Telomeres of HMCs

According to the results of modeling experiments, it is known that the presence of qRRMs in hnRNPs does not ensure the ability of such proteins to bind to ssDNA [89]. However, it was found that some proteins in the hnRNP-F/H family, each containing three qRRMs, could bind telomeric RNA and prevent G-quadruplex formation in telomeres [40]. It was shown that telomeric RNA is transcribed from the telomeric C-rich strand, giving rise to repeat-containing telomeric transcripts or TERRA, and hnRNP-F is a TERRA-bound protein [200]. Thus, it is reasonable to hypothesize that hnRNP-F and probably other proteins in the hnRNP-F/H family provide their own contributions to the function of telomeres.

HnRNPs containing KH domains have been actively studied in HMCs for more than 20 years, and the multifunctional hnRNP-K has attracted the most attention. Numerous and sometimes contradictory data on hnRNP-K were discussed in recent reviews [201,202]. It is currently believed that this protein participates as an important player in processes associated with carcinogenesis. For example, hnRNP-K was found to be overexpressed in several human cancers, and its aberrant cytoplasmic localization has been associated with a worse prognosis for patients. Moreover, it was revealed that hnRNP-K, as a multifunctional protein, can regulate both oncogenic and tumor suppressive pathways through the remodeling of chromatin and alteration of different DNA-, RNA-, and protein-mediated activities (for example, [203,204]).

There are some publications proposing several possible mechanisms of hnRNP-K’s involvement in telomere function. It was noted that hnRNP-K is able to recognize and bind to the C-rich strand [namely, d(CCCTAA)n repeats] of vertebrate telomeres [205,206,207]. It has also been shown that, in HMCs, hnRNP-K (like hnRNP-D) is involved in the regulation of hTERT promoter activity [37]. According to these authors, these similar hTERT promoter activities can significantly increase the synthesis of telomerase and ensure the immortalization of HMCs. In addition, hnRNP-K (as a transcription factor) is involved in the G-quadruplex-mediated regulation of the gene expression of the telomere binding protein POT1 [112].

There is limited information about the possible roles of two other hnRNPs containing three KH domains (hnRNP-E1, hnRNP-E2) in telomere maintenance in HMCs. It has been reported that hnRNP-E1 shows remarkable specificity for binding to the telomeric d(CCCTAA)n repeated motif [206]. The observed specific interactions of the KH1 domain in hnRNP-E2 with telomeric DNA and telomerase RNA provide grounds for assuming that hnRNP-E1 and hnRNP-E2 may participate in mechanisms involved in the regulation of telomere/telomerase functions [207,208].

Thus, KH-containing hnRNPs might play some roles in telomere maintenance in HMCs; however, the presented data and hypotheses clearly need confirmation and clarification.

It is known that the amino acid sequence of hnRNP-U (synonym: SAF-A) has neither RRM nor qRRM nor KH domains, but this protein contains several RGG motifs that are involved in the formation of the special arginine/glycine-rich domain (RGG/RG) [Q00839 UniProt]. So far, direct data on the participation of hnRNP-U in telomere elongation processes in HMCs have been presented by only Fu and Collins (2007) [2]. The authors of that publication drew an important conclusion: “endogenous human telomerase complexes are more heterogeneous than those of single-celled eukaryotes, have predominantly shared rather than telomerase-specific proteins, and make numerous regulatory interactions.” 

It was later shown that the RGG/RG domain is able to recognize G-quadruplex structures [209], which suggests possible binding of hnRNP-U with the corresponding telomere components. 

In general, it seems that some hnRNPs without RRM domains can interact with various components of telomeres. However, the functional significance of these interactions and their role in maintaining telomeres require additional studies.

## 4. Discussion

The current data on the significant diversity of hnRNPs create certain difficulties when describing individual proteins within the framework of the traditional alphabetical classification, although it may be crucial when studying their involvement in certain physiological and pathological processes, particularly those related to telomere function in HMCs.

In accordance with the alphabetical classification, to denote genes that are considered closely related, additional digital indices need to be used (for example, *HNRNPA1, HNRNPA2B1, HNRNPA3, HNRNPA0, HNRPAB, HNRNPA1L2*). This approach is also used to name the corresponding protein products. Since the expression of most genes encoding hnRNPs occurs with alternative splicing, several transcripts are formed from a single gene. As a result, during the synthesis of the protein products of genes, different proteins can be formed, though they are very similar in electrophoretic and other properties. Confusion sometimes arises when using the traditional alphabetical classification. For example, the symbol “B” was recommended for one of the products of the *HNRNPA1* gene and also for one of the products of the *HNRNPA2B1* gene. There are even cases when the products of different genes receive almost the same designation using the alphabetical classification. For example, in their review, He and Smith (2009) [61] indicated that the same symbol, hnRNP-A2, can be used to designate the alternative splicing products of two different genes, *HNRNPA2B1* and *HNRNPA3*. Thus, in the designation of various human hnRNPs, it is important to supplement the symbols corresponding to the traditional classification with more universal symbols, such as those used in the UniProt database, and to take into account current genomic and transcriptome data.

At the same time, a number of genes and their corresponding hnRNPs have alternative and quite actively used names (e.g., hnRNP-E1, hnRNP-E2, poly(rC)-binding proteins PCBP1 and PCBP2, for example, [210]). Occasionally, one hnRNP may have many different names and abbreviations. For instance, the following names are used as synonyms for the designation of heterogeneous nuclear ribonucleoprotein P2 (hnRNP P2): RNA-binding protein FUS, 75 kDa DNA-pairing protein, translocated in liposarcoma protein, G-quadruplex telomere DNA- and TERRA-binding protein TLS/FUS, FUS/TLS (fused in sarcoma/translocated in sarcoma), etc. [115,197,198,211,212]; P35637 UniProt. Moreover, the FUS symbol is recognized as being official for the gene encoding hnRNP-P2 (according to Gene NCBI ID: 2521, updated on 12 August 2018), although, in the corresponding description, it is noted that it is “*also known as: TLS; ALS6; ETM4; FUS1; POMP75; HNRNPP2. This gene encodes a multifunctional protein component of the heterogeneous nuclear ribonucleoprotein (hnRNP) complex*.” Despite the fact that information about the RNA-binding protein FUS (P35637) is contained in the UniProt database, the symbols hnRNP-P2 and HNRNPP2 are not used in the corresponding annotation.

Creating a convenient numbering system for various hnRNPs can apparently be considered an urgent task, whose solution will enable the optimization of ongoing research. Individual designations of hnRNPs (reflecting information on phosphorylation and other post-translational modifications) for which an ability to interact with telomeres has been established could significantly contribute to the development of ideas concerning possible mechanisms of their influence on telomerase function [73,152,153].

Considering the likelihood that hnRNPs with similar structures (proteoforms or products of related genes) have similar properties, it is reasonable to analyze the materials on their interactions with telomeres while taking into account the content of different RNA-binding domains in hnRNPs.

## 5. Conclusions

Currently, the different domains and motifs that determine the ability of hnRNPs to bind various nucleotide sequences are established. Special emphasis is placed on hnRNPs that contain RRM domains, which permit recognition of and binding to d(TTAGGG) repeats in a terminal 3′ single-stranded overhang of telomeric DNA. In particular, hnRNP-A1 and hnRNP-A2, which have two structurally similar RRM domains as well as several RGG motifs, are the most convincing participants in telomere maintenance. Moreover, these functions are preserved in an artificial protein (UP-1, produced from hnRNP-A1) that lacks RGG motifs but contains RRM domains. Additionally, it has been shown that hnRNP-A18, which contains only one RRM that is structurally similar to the hnRNP-A1 and hnRNP-A2 domains, is also involved in telomere maintenance in HMCs. Thus, it can be concluded that these and other hnRNPs with RRM domains similar in structure to the RRM domains of hnRNP-A1 are of particular interest for further research as potential participants in telomere maintenance.

The data on interactions between telomeres and hnRNP-A1 as well as other members of the hnRNP-A/B family reveal several important circumstances. First, hnRNP-A1 and hnRNP-A2 are expressed in considerable excess compared with telomerase, as well as the number of telomeres in HMCs. Second, there are different functional overlaps among the main members of the hnRNP-A/B family. Third, members of the hnRNP-A/B family exist in HMCs not only as products of the expression of several related genes but also as various proteoforms, which form due to alternative splicing and post-translational modifications. These circumstances should be taken into account when forming ideas concerning the molecular mechanisms of the influence of hnRNP-A1, as well as other members of the hnRNP-A/B family, on the dynamics of telomeres in HMCs.

Evidence of interactions between telomeres and hnRNPs that do not contain RRM domains is rather limited. However, it was shown that hnRNP-K is able to recognize and bind the C-rich strand [namely, d(CCCTAA)n repeats] of vertebrate telomeres. The significance of such interactions in HMCs still needs to be detailed.

Thus, from further research on hnRNP involvement in telomere function, one can expect the development of views on the role of hnRNPs in carcinogenesis and, on this basis, the creation of new methods to suppress the proliferative activity of HMCs.

## Figures and Tables

**Figure 1 ijms-20-00745-f001:**
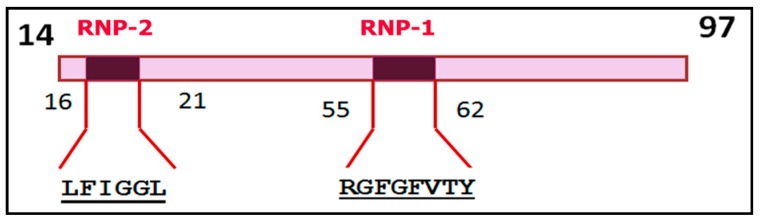
Schematic representation of the RRM1 (RNA recognition motif) domain of heterogeneous nuclear ribonucleoprotein A1 (hnRNP-A1) with the RNP-1 and RNP-2 submotifs [57] and P09651 UniProt. The numbers correspond to the positions of amino acid residues in the hnRNP-A1 polypeptide chain.

**Figure 2 ijms-20-00745-f002:**
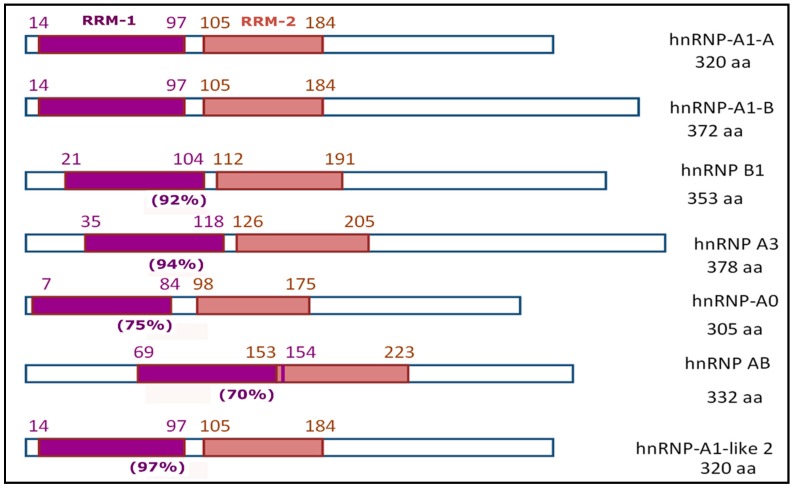
General schemes of the structure of proteins belonging to the hnRNP-A/B family (including two alternative splicing products of the *HNRNPA1* gene) according to UniProt; [45,61]. In parentheses are the homology estimates of the RRM-1 domains compared with the RRM-1 domain in hnRNP-A1. Hereinafter, all calculations are made using the Needleman–Wunsch Global Align Protein Sequences program (via https://www.ncbi.nlm.nih.gov/).

**Figure 3 ijms-20-00745-f003:**
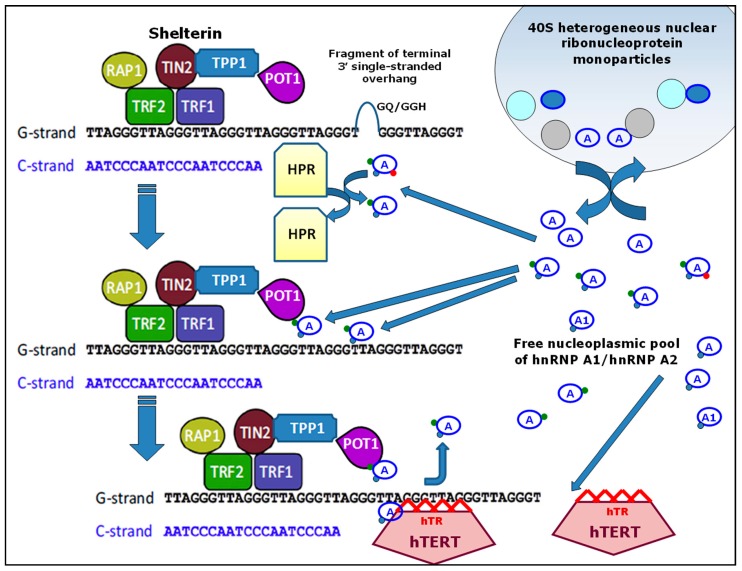
A hypothetical scheme of the possible dynamic relationship between hnRNP-A1/hnRNP-A2 and some components of telomeres. Molecules of hnRNP-A1 or hnRNP-A2 are shown by the symbol A in blue ovals; red, green, and blue points indicate various post-translational modifications. HPR—hypothetical protein regulator; GQ/GGH—G-quadruplex or G-G hairpin structures; the remaining designations (abbreviations) and commentary are presented in the text.

**Table 1 ijms-20-00745-t001:** The amino acid sequences of the RNP-1 and RNP-2 submotifs in members of the hnRNP-A/B family (according to UniProt). Numbers indicate the positions of amino acid residues in the corresponding polypeptide chains. Highly conserved amino acid residues, which are identical to the residues in hnRNP-A1, are highlighted in gray.

Members of hnRNP-A/B Family	RRM-1		RRM-2	
	RNP-1	RNP-2	RNP-1	RNP-2
hnRNP-A1	**55** **62** RGFGFVTY	**16** **21** LFIGGL	**116** **153** RGFAFVTF	**107** **112** IFVGGI
hnRNP-B1	**62** **69** RGFGFVTF	**23** **28** LFIGGL	**153** **160** RGFGFVTF	**114** **119** **LFVGGI**
hnRNP-A3	**76** **83** RGFGFVTY	**37** **42** LFIGGL	**167** **174** RGFAFVTF	**128** **133** IFVGGI
hnRNP-A0	**48** **55** RCFGFVTY	**9** **14** LFIGGL	**139** **146** RGFAFVTF	**100** **105** **LFVGGL**
hnRNP-A1-like 2	**55** **62** RGFGFVTY	**16** **21** LFIGGL	**156** **163** RGFAFVTF	**107** **112** IFVGGI
hnRNP-AB	**110** **117** RGFGFILF	**71** **76** **MFVGGL**	**195** **202** RGFVFITF	**155** **160** IFVGGL

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
