# Peer review of "Heterogeneous Nuclear Ribonucleoproteins Involved in the Functioning of Telomeres in Malignant Cells"

_ijms, 2019, doi:10.3390/ijms20030745_

Round 1

Reviewer 1 Report

In this manuscript, authors summarized the studies of hnRNPs structurally different proteins and their relationship with telomerase complex activity in malignant cells.

Authors summarized abundant publications of hnRNPs genes and proteins, and provided lots of information for other researchers. However, I considered that authors should organize the manuscript better.

In general, authors separates the manuscript into three sections.

For example, in the second section, 2 General properties of several main human hnRNPs involved in telomere functions. Authors should summarize the content into several sub-titles, such as to separate this information by the structure or sub-motif.

For the same reason, in the third section, authors made a statement of the relationship between hnRNPs and telomeres. Authors did it better to separate this section into three parts by the hnRNPs structure, but I considered that authors could separate more detail to make the paper more easy to be read. For example, in 3.1 Members of hnRNP-A/B family and telomers of HMCs. Authors could group the content by their binding proteins of telomerase complex.

Overall, I considered this manuscript was very good and could give more useful clues to researchers in related field.

Author Response

Dear reviewer,

I and my colleagues are thanks You for the great work done and positive report about our manuscript ”Heterogeneous nuclear ribonucleoproteins, involved in the functioning of malignant cells telomeres”.

We have revised our manuscript according to the comments. The content of our article has been restructured and additional sub-titles have been added in section 2 and section 3 that highlighted in the text by green marker.

I and my colleagues warmest wish you for a happy celebration and a wonderful new year!

Best Regards,

S
ergey S. Shishkin

Reviewer 2 Report

This is a very detailed and conscientious review about heterogeneous nuclear ribonucleoproteins and their potential or already established interactions with telomeres in malignant cells.

Most sentences and English grammar are ok, but language has to be amended and corrected by a native speaker. Some sentences are clumsy or incorrect in grammar and thus difficult to understand..

This already starts with the heading which should be: "Heterogeneous nuclear ribonucleoproteins involved in the functioning of telomeres in malignant cells".

The first half of the review is an exclusive description of coding and structure of all the different hnRNP proteins while only the second half deals with their interaction with telomeres. I am wondering whether the first part can be a bit streamlined and a bit shortened  since in my view it is not immediately connected to the functional interaction with telomeres. Also, a better rational should be given why the authors emphasise the role in cancer cells while the proteins certainly also interact with telomeres in normal cells.

In addition, quite a bit is known about the important interaction of hnRNPA1 with TERRA and its regulation of telomere length and telomerase activity. This should be described in more details since it is essential for the topic of the review (see Flynn et al., 2011, Redon et al., 2013, Yamada et al., 2016).

Regarding the various language issues: please avoid the term "modern" and replace with "current" or "recent".

I don't really like the 1st sentence of the introduction starting with "The development of modern opinions about the functioning of the genomes in human malignant cells...". I don't think the review should deal with "opinions" or their development but with facts and evidence. I am also not sure that "genomes are functioning". Regarding telomeres and "the role of a number of proteins" this is very general and does not contain any real information.. Please try to amend this sentence.

Other language issue are "mechanisms providing carcinogenesis" on page 1 last line,"Providing" should be substituted with "promoting".  Page 9: "Violation of... phosphorylation" should be substituted with "failure of... phosphorylation". Page 11 under conclusion: the sentence containing"... family it was enalble note several important circumstamces" is completely incomprehendable to me-what exactly want the authors say here? Also in conclusions: ...”with lacked RGG motifs” should be “lacking RGG motifs”. These are only some selescted examples. In general, the use of articles like "the/a", times (past or presence) as well as plural and singular have to be extensively amended.

I am sure that after these corrections it could be a rather interesting review.

Author Response

Dear reviewer,

I and my colleagues thanks You for the great work done and positive report about our manuscript ”Heterogeneous nuclear ribonucleoproteins, involved in the functioning of malignant cells telomeres”.

We have accepted all your comments and made appropriate corrections to the manuscript.

1. The content of our article has been restructured and additional sub-titles have been added in section 2 and section 3 that highlighted in the text by green marker.

2. The section 2 of the manuscript is shortened, in particular table 2 is removed and the discussion of classification questions is moved to section 4 (Conclusions). In this regard, the numbering of the cited literature has been reworked.

3. The articles of Flynn et al., (2011), Redon et al., (2013), Yamada et al., (2016) have included in references (numbers 9, 99 and 123 consequently) and cited in several sections of manuscript. Data from these articles about the interaction with hnRNPA1, TERRA as well as on the participation of this protein in the regulation of telomere length and activity of the telomeres have included in the section 3, lines: 328-330, 349-354.

4. Some data about the presence and roles of hnRNPs in normal cells as well as its possibility to interact with telomeres in normal cells introduced in section 3, lines; 445-454

5. We checked our manuscript and amended some sentences that were incorrect in grammar and did not contain any real information as well as amended all identified grammatical inconsistencies. We agree that the review should not deal with "opinions" or their development and deleted the first sentence of the introduction. I and my colleagues determined that the sentence "The data collected on the interactions between telosomes and hnRNP-A1 as well as other members of the hnRNP-A/B family it was enable note several important circumstances." in conclusion is aslo quite incomprehendable and should be corrected. Thus, we decided the sentence “In conclusion of this review it should be noted that interactions between telosomes and members of the hnRNP-A/B family runs at several important circumstances” The "...lacked RGG motifs... " has been replaced by “lacking RGG motifs”.

In revised article we avoid the term "modern" and replaced with "current"; the word "providing" has been substituted with "promoting" in introduction section;. the phrase "violation of... phosphorylation" on page 9 also has been revised, as You recommended.  

I and my colleagues warmest wish you for a happy celebration and a wonderful new year!

Best Regards,

S
ergey S. Shishkin

Round 2

Reviewer 2 Report

After the 1st review round the manuscript has been substantially improved and is now better organised. More information was added for the relevant topic which is the interaction of hnRNPs with telomeres. However, the first 6 pages still deal almost exclusively with structural issues of hnRNPs and various domains without any obvious relation to the second, more functional part on pages 7-14. The authors should better explain what the functional biological significance of the different hnRNP domains-in general as well as at telomeres.

In addition. there are still some issues that have to be addressed and I list them below:

Title:  while the response letters claims that the title has been changed it hasn't and the grammar of "malignant cells telomeres" is still wrong and has been translated from Russian. IF anything, then it has to be "cell's telomeres" although it is still very poor English.

Although the authors claim that a native speaker has corrected the manuscript text, there are still hundreds of grammar and language mistakes which I list for the authors, so that they can correct them eventually. I will list all the hundreds of other grammar and language errors at the end. I am not sure how careful the native English speaker has corrected the manuscript. Often either praedicates or subjects are missing from the sentence or the sentence contain so many sub-sentences that the original sense gets lost and grammar is wrong.

A proper introduction into telomeres and telomerase has to be given in the beginning of the review since this is only provided half through the review on page 7 when telomeres, telosomes and telomerase have been mentioned repeatedly. Importantly, “telosomes” and what their difference is to telomeres have never been described. This is important to avoid confusion.

Here are some details about mentioning of telomeres and telomerase before their proper introduction: Although "telomere function" is already mentioned in heading 2 on page 1, telomeres are not introduced and mentioned  until page 7. Under heading 2 only an occasional very general statement about telomeres is included which does not supply any meaningful information for the reader. Examples are line 72 on page 2 stating: "..hnRNPs have the ability to bind both DNA and RNA which is of fundamental importance for their participation in interactions with telomeres." No explanation how and why this is important for readers not familiar with telomeres. On page 5 line 166 again a statement that hnRNP isoforms and changes might have effects on telomere stability-no further details about mechanisms or experiments/studies are given. Although in line 200 "telomerase" is mentioned, it has not been introduced and described before-please ADD! Likewise, in line 217 "telomeric RNA" is mentioned, yet TERRA or the telomere as such has not been introduced yet. How should the reader not from the telomere filed understand this?

Another mention on potential interactions with telomeres in line 172. In line 200 telomerase is mentioned but again, this enzyme, its structure and significance for telomeres has NOT been introduced yet at all at that stage, so the reader  not familiar with this subject will not understand it. In particular, the importance of telomerase activity in cancer cells in contrast to most human somatic cells that do NOT have the enzyme (just the RNA hTR, not hTERT) should be emphasised for readers not from the field. Consequently, it is also important to mention that ONLY in cancer cells telomerase might be a part of the "telosome" while in most human somatic cells it is not.

Often very general and not informative statements are given, for example on page 4 under line 137/8 authors write: “..that even LEAD to pathology, for example”-and just give some references. you cannot just give references here, but you have to describe what pathology has been shown- is this cancer in vivo? Since cell lines normally don’t have “pathologies”... Please specify

The conclusion which was a reasonable half page before has now been extended and is with 1.5 pages overlong. While the second part correctly summarises what the review said about interactions between telomeres and some hnRNPs, the first part now describes problems with hnRNP’s nomenclature that have not been mentioned before, so this part has to be shifted elsewhere, perhaps to the general introduction or as a separate sub-heading since a conclusion should summarise in brief the content of the review and not introduce new aspects.

Line 346: I think the authors mean “R-cycles” here and I am not sure that they contain “G-quadruplexes”. Please double-check since in my understanding quadruplexes do NOT form R-loops/cycles. Please define these R-loops/cucles correctly.

Minor issues:

line 225 what is Poly(rC)-binding? Could you please explain these special terms and abbrevations!

line 241: Why are KH domains the “most common intramolecular structures”?

256: what do you mean here with “... this shortening eliminates a particular enzyme?” And what enzyme do you refer to? This statement is completely incomprehendable. Please correct/amend.

261: some mouse telomeres can be up to 80 kB!

264: It is not correct to state that telomerase always adds around 60 nucleotides (10 hexanucleotide repeats. This depends on the processivity of the enzyme, the species, cell type etc. It can have very low processivity and only add 1 repeat in mouse cells up to many more in different cancer cells. So please remove this wrong statement.

265: it is also not correct that telomerase is necessarily a part of telosomes-this is only tru in telomerase positive cells and only when telomerase is active at the telomeres, while most of the time it is sequestered away from telomeres. Thus, I would be very careful to call telomerase a part of the telosome in general. All depends again on the cell type and cell cycle phase etc.

302: the REPLICATION FACTOR A (RPA) is predominantly involved in DNA replication (as the name states. In addition, it participates in homologous recombination which is just one form of DNA repair...

303: the statement that RPA interacts "with  components IN THE BODY" is really completely hollow and meaningless-what are these components? Please avoid such general statements which do not give any information and be more specific!

Line 418: Please ex[lain how the described hypothetic protein interactions would be able to increase telomerase enzyme activity which is mainly regulated at the transcriptional level and by posttranscriptional modification of hTERT. Known knowledge about telomerase activity regulation should be taken into account.

Lines 431-434: Pot binds to ss telomeric DNA (overhang) only OUTSIDE telomerase reaction. In order to uncap telomeres for telomerase action POT1 is DISSOCIATED from the telomere. This is all known and well described, so please correct this wrong statement even though it is a hypothesis. The latter should not contradict already known facts.

Fugure 3: The above also applies to the scheme in fig 3 which shows a binding of TRF1 and 2 to ss DNA, but the proteins bind exclusively to ds DNA. Again: Pot1 is shown bound to the ss overhang upon telomerase acyion which is WRONG since the former it is sequestered away in order to allow telomerase to bind to the opened G-overhang. Please correct and include relevant literature on the correct facts.

Grammar and language errors:

Title “malignant cells telomeres” is wrong grammar. Better: telomeres in malignant cells

Abstract line 16: “interacts with TELOMERE or TELOMERIC DNA”

Line 18: “TELOMERE length”, NO genitive like in Russian! “Accordingly, these proteins ARE considered AS possible...”

Line 19/20: “.. we discuss the results OF STUDIES...

Introduction line 25: “Over the past decade STUDIES...” line 26 “HAVE constituted...”(related to “studies”=plural! Also: “.. a whole fron of research” is not something really used in English...

Line 30/31: “...and increase OF telomerase activity...”

Line 34/35: “ ...participate in telomere...”

36: “...in telomere functioning”... no article

Line 52 and all other citation: ONLY the surname of an author is used, not his initials.

Lin 65: “HAVE not yet been found...” –relates to genes=plural

69: majority (is singular!) of... CONTAINS...”

Line 84: HnRNPs

Line 102: “...domains OVERLAP” (Plural!)

Line 108: “... lower WITH 34-46%”

112: “are encoded BY five...”

119/120: “Consequently, THE LARGELY positively charged arginine radical..IS formed AT this site...”

121: “... while in the adjacent amino acid residues ONLY PROTONS with a mass of 1D ARE PRESENT instead of radicals.”

126: “...which ARE identical...”

127 “highlighted IN GREY”

130: “...12) OF amino acid...”

137/8: “..that even LEAD to pathology, ...”-

142: “...isoform A1A with A Mm OF 84 kDa” this is a recurrent mistake, please correct everywhere (line 149 and others)155: you talk about “telosomes” but you have NOT yet described what that is!

146” ...COLO-RECTAL cancer....”

147: remove “the” in front of hn-RNPA1, article not required here.

148 please correct to “...as one of the 500 MOST ABUNDANT proteins” there is not such a thing like an “abundance protein”!!!!

165 what means the “(and are becoming)” changes...”? This is not clear, please remove the words in brackets

182: “...SHOWN BY proteomic...”

189: you state “was found in several HMCs”, but provide only ONE example and reference, in this case you should add “such as, for example HL-60 cells and remove the “leukemia cells” from the bracket since you do not explain most other cell lines and readers working in the cancer field know most of the common cell lines.

190: “..is caused by A recent report THAT HNRPDL...”-you should write “hNRNP” consistently with a small “n” and not switch from small to capital letter and back again... Please amend through-out the text.

193: “In contrast to members of THE human...”

195 “.. DO not...”

200 “... IS low...able to interact with THE RNA part...”

209 “.. encode..” (relates to 4 GENES=plural!

211: “THE HnRNP... , as A result”

213: “HnRNPs containing three qRRMs ARE considered MEMBERS ...” no comma required

215: sentence is incomplete”...a role these proteins MIGHT PLAY in...”

217 “[17] and IT IS possible...

238: “... are joint INTO A special protein family.”

241: Why are KH domains the “most common intramolecular structures”?

243: 2.5 HnRNPs with A RGG/RC domain-please remove the 1 bracket!

248 RNAs

253 telomere-PROTEIN relationships

262: “In HMCs there is A special enzyme?

269: “DNA of telomeres IS transcribed..”

270 “TERRA is considered A key component...”

272: “...as well as in MAINTENANCE OF TELOMERE length” NOT chromosome length! This is not relevant here!

273” RNA IS involved..”

276: In particular THE human telosome contains A six-protein...

277 “... which IS COMPRISED OF...”

281: “... and IS involved...”

285 remove “respectively

288: “... OTHERS are SIGNIFICANTLY smaller...”

289: “..Mm of THE total...”

297/8: “... telomeres IN a non-sequence specific manner and BINDS TO single-...”

299”... HAS BEEN noted”

333: “…data WAS obtained.” No article required here.

340: …”capable to UNFOLD…”

342: what is meant with “AN appropriate conclusion was confirmed…”? What conclusion do you mean and why is it “appropriate”? Please amend.

343: “Besides,…”

345: “…confirmed by photochemical…experiments that LOOPS in the…”

347: “…the base of A nucleotide at THE loop of…”

350: “ Redon et al HAVE shown…”

354:”…influencing hnRNPA1 localisation…”

363 please write “in vitro” and “in vivo” in italic

364:” (DNA-PK) PRESENT in…”

366:”…that RNP-A1 phosphorylation…”

375:” phosphorylation of ANOTHER…”

360:” …considered A potential///”

384”…evidenced FOR…”

386: “..expression OF not…”

389: …it can recognise WHAT?-stranded? Single or double? Or what do you mean with “stranded”?

390: “ …isoforms WAS observed…”

398: “members OF the…”

401: “despite THEIR LARGE similarity”

406: “…in human CELLS…”

411: “…the available data SUGGESTS two assumptions: “ continue without new paragraph

412: “…at THE terminal…”

417: “ …are POSSIBLY necessary…”

420: “… regulation may not BE INFLUENCED BY ANY, but only BY SOME….”

422:” …was discussed UNDERTHE aspect of…

425/434/444 “…maintenance of TELOMERES…”

426:”  …are ABLE to displace…”

428:” …COULD interact..”

430:” ..REPEATS would…”

431:” …creation OF conditions…”

433:” will provide the binding…”

434: “ proteoforms PRESENT …” or “that ARE PRESENT”

445:” …members of THE hnRNPs…”

449: “… ARE involved…”

450: “ mediated THE reduction…”

451: “ribonucleoPARTICLE”

452: “ ... normal mortal cells.” Cell LINES are always IMMORTAL. “… thought that members of..”

461: “ …A2 ARE SHOWN BY THE symbol…”

466: “…detected FOR another…”

469: “… telomerase ACTIVITY…”

472: “…CONSIDERED A useful…”

479: “….according to available data…”

480: “ A18) INVOLVES ALTERNATIVE SPLICING…”

481: “..has THE structural…”

484: “…upon binding, THUS PROMOTING maintenance…”

489: “ … in the NUCLEUS…”

491: “ consider them A separate family…”

493: “… members of THE hnRNP…”

496: “ ..may be THE basis…”

500: “ …which IS located at THE 5’ terminus…”

503: “ by AN immunofluorescence method”

504: “..telomere binding…”

505 : “… but ARE not shown…

505/06: “C1/C2 might bind…” please add a dot after “interactions”

509: “…of interaction BETWEEN TELOMERES and the hnRNP-P2 which contains…”

512: “ .. and the gene coding it WAS DESCRIBED BY VARIOUS AUTHORS as oncogenes…”

515: “ contribution TO DNA-RNA binding…within A zinc finger domain PLAYS A MAIN ROLE..”

519: “ …EVIDENCE EMERGED…”

523: “ Information on possible direct and indirect INTERACTIONS OF TELOMERES WITH hnRNPs

529: “…RNA IS transcribed…”

531: “ is A TERRA bound protein”

535: “…attracting A LOT OF attention…”

537:” .player in THE carcinogenesis..”

540: “… protein THAT can regulate…”

553: “…hnRNP E HAS shown…binding TO the telomeric…”

554: ” ..of THE KH1…”

555: “E1 AND hnRNP E2…”

557: “ Thus, KH-CONTAINING…”

559@” …sequence OF…there ARE neither A RRM, nor A qRRM…”

562: “ … in THE telomere…process”

563 :” … only BY…”    “… the authors DREW…”

564: “.. single-cell…”

567: “that suggested…possibility OF binding…

571: ” …should be FURTHER studies IN DETAIL.”

577/8: “ …considered closely related…”

578:” …indices SHOULD be used…”

582: “…be formed, very similar…

587: “… HU and Smith…”

607: “… considered AN URGENT TASK…”

611: “ …about possible mechanisms…”

613:” …existence of hnRNPs similar in structure…

618: “Special emphasis is paid TO…”

623: “ … but CONTAINING…”

630: it is absolutely unclear what”runs at several important circumstances” means. Please use a correct English phrase. This her is not comprehendable!

638/9: “ EVIDENCE…IS rather limited”

642: “ …involving TELOMERE FUNCTIONS….”

Author Response

Dear reviewer 2,

My colleagues and I thank you for the important contribution to our article and many important comments. We accepted all your comments and tried to make corrections in accordance with them. In particular, based on your recommendations we restructured our article and added data about telomeres, telomerase, telosomes and TERRA briefly in introduction.

We apologies for making some mistakes in figure 3 and in corresponding description especially regards for TRF1, TRF2 and POT1. This figure and its description have been revised and supplemented by new references.

All amendments made to the text are highlighted in yellow and discussed in detail below in our point-by-point response.

Our point-by-point responses.

1. Reviewer comment: “However, the first 6 pages still deal almost exclusively with structural issues of hnRNPs and various domains without any obvious relation to the second, more functional part on pages 7-14. The authors should better explain what the functional biological significance of the different hnRNP domains-in general as well as at telomeres”.

Response to the comment: The information about the functional biological significance of the RRM domains of hnRNPs in general as well as at telomeres provided on page 3 (lines 101-116). Moreover, there is an information on the characteristics and functional biological significance of the RRM domains in hnRNP-D (lines 227-231) and hnRNP C1/C2 (lines 247-249) on page 6 of second revision. The functional biological significance of the qRRMs domains and their possible interactions with telomeres are discussed on page 7 (lines 256-259) and lines 266-270, correspondingly. Additional data about the functional biological significance of KH-domains is given on page 7 (lines 276-279). Information on the functional biological significance of RGG/RG domains is located on page 8 (lines 301-305). All added data is justified by references which also included in list of the literature.

The parts 2.2, 2.4 and 2.5 of section 2 are slightly shortened and several references have been removed.

2. Reviewer comment: “Title: while the response letters claims, that the title has been changed it hasn't and the grammar of "malignant cells telomeres" is still wrong and has been translated from Russian. IF anything, then it has to be "cell's telomeres" although it is still very poor English.

Response to the comment: The title of the article has been changed as it proposed after the first round revision.

3. Reviewer comment: “Although the authors claim that a native speaker has corrected the manuscript text, there are still hundreds of grammar and language mistakes which I list for the authors, so that they can correct them eventually. I will list all the hundreds of other grammar and language errors at the end. I am not sure how careful the native English speaker has corrected the manuscript. Often either praedicates or subjects are missing from the sentence or the sentence contain so many sub-sentences that the original sense gets lost and grammar is wrong”.

Response to the comment: We also thank you very much for the pointing to the number of additional grammatical errors in manuscript after first round of revision. To improve the language of our article we used the English editing service provided by MDPI and received the English editing certificate that the manuscript text has been checked by native English speaking editors.

4. Reviewer comment: “A proper introduction into telomeres and telomerase has to be given in the beginning of the review since this is only provided half through the review on page 7 when telomeres, telosomes and telomerase have been mentioned repeatedly. Importantly, “telosomes” and what their difference is to telomeres have never been described. This is important to avoid confusion”.

Response to the comment: Some background information about terminology of telomere, telomerase and telosomes with several references is included in introduction section and partly some information is moved from section 3.1 of first round revision (in second round revision - lines 25-59). Accordingly, in second round revision the all references have been renumbered and new added references shown yellow marker in the literature list.

5. Reviewer comment: “Here are some details about mentioning of telomeres and telomerase before their proper introduction: Although "telomere function" is already mentioned in heading 2 on page 1, telomeres are not introduced and mentioned until page 7 Under heading 2 only an occasional very general statement about telomeres is included which does not supply any meaningful information for the reader”.

Response to the comment: The short description of structure and functions of telomere is included in introduction section (lines 40-48 and 53-59, correspondingly). The corresponding references have been added in the literature list. 

6. Reviewer comment: “Examples are line 72 on page 2 stating: "..hnRNPs have the ability to bind both DNA and RNA which is of fundamental importance for their participation in interactions with telomeres." No explanation how and why this is important for readers not familiar with telomeres.

Response to the comment: Relevant explanations given on page 3 of the second revision (lines 110-116).

7. Reviewer comment: “On page 5 line 166 again a statement that hnRNP isoforms and changes might have effects on telomere stability-no further details about mechanisms or experiments/studies are given.

Response to the comment: Data of experimental studies providing explanations for the possible different hnRNPs isoforms effects on telomere stability is cited on page 6 (lines 208-210).

8. Reviewer comment: Although in line 200 "telomerase" is mentioned, it has not been introduced and described before-please ADD! Likewise, in line 217 "telomeric RNA" is mentioned, yet TERRA or the telomere as such has not been introduced yet. How should the reader not from the telomere filed understand this?”

Response to the comment: The brief information about telomerase and TERRA have been moved from section 3.1, expanded of new data and included in introduction section (lines 27-30 and 57-59, correspondingly).

9. Reviewer comment: “Another mention on potential interactions with telomeres in line 172 In line 200 telomerase is mentioned but again, this enzyme, its structure and significance for telomeres has NOT been introduced yet at all at that stage, so the reader not familiar with this subject will not understand it. In particular, the importance of telomerase activity in cancer cells in contrast to most human somatic cells that do NOT have the enzyme (just the RNA hTR, not hTERT) should be emphasised for readers not from the field. Consequently, it is also important to mention that ONLY in cancer cells telomerase might be a part of the "telosome" while in most human somatic cells it is not.

Response to the comment: The definitions of hTERT and hTR as well as several peculiarities of telomerase functioning in malignant cell in comparison with normal somatic cells have been included in introduction section (lines 27-30 and 46-52, correspondingly). The literature list has been supplemented by additional references. Moreover more data about hTERT and hTR in human malignant cells have been add to the manuscript and corresponding reference added (lines 324-328).

10. Reviewer comment: “Often very general and not informative statements are given, for example on page 4 under line 137/8 authors write: “..that even LEAD to pathology, for example”-and just give some references. you cannot just give references here, but you have to describe what pathology has been shown- is this cancer in vivo? Since cell lines normally don’t have “pathologies”... Please specify.

Response to the comment: A clarification of that issue has been made in second round revision (lines 178-181).

11. Reviewer comment: The conclusion which was a reasonable half page before has now been extended and is with 1.5 pages overlong. While the second part correctly summarises what the review said about interactions between telomeres and some hnRNPs, the first part now describes problems with hnRNP’s nomenclature that have not been mentioned before, so this part has to be shifted elsewhere, perhaps to the general introduction or as a separate sub-heading since a conclusion should summarise in brief the content of the review and not introduce new aspects”.

Response to the comment: In second revision, the conclusion section has been shortened. The part of text which describes problems with hnRNP’s nomenclature has been formed as a separate section of discussion.

12. Reviewer comment:” Line 346: I think the authors mean “R-cycles” here and I am not sure that they contain “G- quadruplexes”. Please double-check since in my understanding quadruplexes do NOT form R- loops/cycles. Please define these R-loops/cucles correctly.

Dear reviewer, unfortunately we weren’t able to find the term “R-cycles” on line 346 and other text in first round revision. On line 346 and further the terms “loops in the telomere RNA G-quadruplex” have been used. These terms taken from the report of Liu X et al. 2017; 2018 and others.

We also amended all incorrections which have been indicated as minor issues in second round review:

1. Reviewer comment: “line 225 what is Poly(rC)-binding? Could you please explain these special terms and abbrevations!

Response to the comment: Some explanations have been added in subsection 2.4 (lines 275-279).

2. Reviewer comment: “line 241: Why are KH domains the “most common intramolecular structures”?”

Response to the comment: The phrase “Accordingly, the accumulated information gives reason to consider KH domains as one of the most common intramolecular structures, which bind RNA or ssDNA and highly conserved from bacteria to mammals [73; 71; 68; 72] (lines 240-242 in text after the first round revision) has been deleted.

3. Reviewer comment: “what do you mean here with “... this shortening eliminates a particular enzyme?” And what enzyme do you refer to? This statement is completely incomprehendable. Please correct/amend”.

Response to the comment: This statement has been corrected. In revised manuscript, it has been amended on “In 1971, A. Olovnikov hypothesized that with each cell division, the DNA at chromosomal ends (telomeres) is slightly shortened, and in some (cancer) cells, this shortening eliminates a special enzyme (called telomerase). Confirming this proposed hypothesis has generated considerable interest in studying telomere structure and dynamics [1–7]”. (lines 25-28)

4. Reviewer comment: “some mouse telomeres can be up to 80 kB!”

Response to the comment: The phrase from line 261 of the first round revision has been deleted in second round revision.

5. Reviewer comment: “It is not correct to state that telomerase always adds around 60 nucleotides (10 hexanucleotide repeats). This depends on the processivity of the enzyme, the species, cell type etc. It can have very low processivity and only add 1 repeat in mouse cells up to many more in different cancer cells. So please remove this wrong statement”.

Response to the comment: This statement has been deleted.

6. Reviewer comment: “it is also not correct that telomerase is necessarily a part of telosomes-this is only tru in telomerase positive cells and only when telomerase is active at the telomeres, while most of the time it is sequestered away from telomeres. Thus, I would be very careful to call telomerase a part of the telosome in general. All depends again on the cell type and cell cycle phase etc”.

Response to the comment: The phrase “In telosomes, the telomerase holoenzyme, at minimum, consists of the reverse transcriptase (hTERT) and an RNA component (hTR) has been corrected.

In second round revision new variant of this phrase has been included in introduction section. The statement telomerase is a part of telosomes” no more used throughout the text.

7. Reviewer comment: “the REPLICATION FACTOR A (RPA) is predominantly involved in DNA replication (as the name states. In addition, it participates in homologous recombination which is just one form of DNA repair...”

Response to the comment: The phrase about participation RPA in control of DNA repair (line 261 of the first round revision) has been changed on “RPA involved in DNA metabolism” (line 369 of the second round revision)

8. Reviewer comment: “the statement that RPA interacts "with components IN THE BODY" is really completely hollow and meaningless-what are these components? Please avoid such general statements which do not give any information and be more specific!”

Response to the comment: This statement has been removed. A phrase RPA …   …may also play a role in telomere maintenance by interaction with some telomere components, including shelterin (lines 370-371) has been added.

9. Reviewer comment: “Line 418: Please explain how the described hypothetic protein interactions would be able to increase telomerase enzyme activity which is mainly regulated at the transcriptional level and by posttranscriptional modification of hTERT. Known knowledge about telomerase activity regulation should be taken into account”.

Response to the comment: The text “increase telomerase enzyme activity” has been removed. The quote from publication of Redon et al (2013) [143] has been added. In this report, it has been noted that hnRNPA1 has no notable direct effects on telomerase catalytic activity (lines 498-499). However, there is a data about allosteric regulation of telomerase activity and relevant short information has been included in manuscript (lines 499-502). We also have noted that the allosteric effects might be considered as possible mechanism of described hypothetic protein interactions would be able to increase telomerase enzyme activity (lines 501-502).

Other data about possible mechanisms of influence of hnRNPA1 and hnRNPA2 interactions with telomere on telomerase functioning is given on lines 490-494 (“stimulates telomere elongation through unwinding of a G-quadruplex or G-G hairpin structures”). The additional information about telomerase activity regulation by alternative splicing with corresponding references has been included (lines 503-505).

10. Reviewer comment: “Lines 431-434: Pot binds to ss telomeric DNA (overhang) only OUTSIDE telomerase reaction. In order to uncap telomeres for telomerase action POT1 is DISSOCIATED from the telomere. This is all known and well described, so please correct this wrong statement even though it is a hypothesis. The latter should not contradict already known facts”.

Response to the comment: A phrase “…would be possible after the binding of POT1 to telomeric ssDNA…” (lines 431- 434 in text after the first round revision) has been removed. The data concerning binding POT1 and telomerase to telomeric ssDNA and their competitive inhibition of each other has been involved in the section 3 of manuscript (lines 327-328; 511-515). In the light of this the figure 3 and its description have been redacted.

11. Reviewer comment: “Fugure 3: The above also applies to the scheme in fig 3 which shows a binding of TRF1 and 2 to ss DNA, but the proteins bind exclusively to ds DNA. Again: Pot1 is shown bound to the ss overhang upon telomerase acyion which is WRONG since the former it is sequestered away in order to allow telomerase to bind to the opened G-overhang. Please correct and include relevant literature on the correct facts”.

Response to the comment: We are sorry for the mistakes in figure 3. On amended figure the TRF1 and 2 binding exclusively to ds DNA have shown. The additional stage has been added for the detalisation of preparation phase of telomerase reaction. This stage shows the liberation ss G-overhang to allow telomerase to bind to the opened G-overhang. In figure caption and description, some relevant and appropriate references have been included. 

All grammar issues have been corrected by us and IJMS English editing service.

Once again, we wish particularly to thank for your large reviewer work and very important comments

With best regards,

Sergey S. Shiskin

Round 3

Reviewer 2 Report

The authors took the majority of my comments on board, corrected mistakes and improved language, there are only very few minor mistakes such as on page 3 line 113: which should be either "telomere functions are changing" or "telomere functioning is changing".

The figure legend for fig 3 has to be moved up underneath the figure. .